# BGP-15 Protects against Doxorubicin-Induced Cell Toxicity via Enhanced Mitochondrial Function

**DOI:** 10.3390/ijms24065269

**Published:** 2023-03-09

**Authors:** Alexandra Gyongyosi, Nikolett Csaki, Agota Peto, Kitti Szoke, Ferenc Fenyvesi, Ildiko Bacskay, Istvan Lekli

**Affiliations:** 1Department of Pharmacology, Faculty of Pharmacy, University of Debrecen, 4032 Debrecen, Hungary; 2Healthcare Industry Institute, University of Debrecen, 4032 Debrecen, Hungary; 3Department of Pharmaceutical Technology, Faculty of Pharmacy, University of Debrecen, 4032 Debrecen, Hungary

**Keywords:** cardiomyocytes, doxorubicin-induced cardiotoxicity, BGP-15, oxidative stress, autophagy, apoptosis, mitochondrial dysfunction

## Abstract

Doxorubicin (DOX) is an efficacious and commonly used chemotherapeutic agent. However, its clinical use is limited due to dose-dependent cardiotoxicity. Several mechanisms have been proposed to play a role in DOX-induced cardiotoxicity, such as free radical generation, oxidative stress, mitochondrial dysfunction, altered apoptosis, and autophagy dysregulation. BGP-15 has a wide range of cytoprotective effects, including mitochondrial protection, but up to now, there is no information about any of its beneficial effects on DOX-induced cardiotoxicity. In this study, we investigated whether the protective effects of BGP-15 pretreatment are predominantly via preserving mitochondrial function, reducing mitochondrial ROS production, and if it has an influence on autophagy processes. H9c2 cardiomyocytes were pretreated with 50 μM of BGP-15 prior to different concentrations (0.1; 1; 3 μM) of DOX exposure. We found that BGP-15 pretreatment significantly improved the cell viability after 12 and 24 h DOX exposure. BGP-15 ameliorated lactate dehydrogenase (LDH) release and cell apoptosis induced by DOX. Additionally, BGP-15 pretreatment attenuated the level of mitochondrial oxidative stress and the loss of mitochondrial membrane potential. Moreover, BGP-15 further slightly modulated the autophagic flux, which was measurably decreased by DOX treatment. Hence, our findings clearly revealed that BGP-15 might be a promising agent for alleviating the cardiotoxicity of DOX. This critical mechanism appears to be given by the protective effect of BGP-15 on mitochondria.

## 1. Introduction

Doxorubicin (DOX) is a potent chemotherapeutic agent widely used to treat a variety of cancers [1]. The clinical use of DOX has been associated with cumulative, dose-dependent cardiotoxicity, while off-target drug toxicity is associated with oxidative stress that involves the development of heart failure [2]. The mechanisms of DOX-induced toxicity have not been clearly elucidated, but are known to involve, at least in part, mitochondrial dysfunction, leading to an increased generation of intracellular ROS, oxidative stress, and apoptosis [3,4]. Thus, DOX cardiotoxicity is closely associated with mitochondrial injury, which is characterized by iron overload and an early loss of mitochondrial membrane potential (MMP) followed by dysregulation of the mitochondrial quality control mechanism [5]. Additionally, DOX activated apoptosis, due to an imbalance in oxidant and anti-oxidant. Therefore, many pathways might be speculated to be responsible for apoptosis induction and there may be cross-talk between these various pathways, including the mitochondrial pathway through caspase-3 activation [6].

More recently, it has been suggested that dysregulation of autophagy may also play a contributing role in DOX-induced cardiotoxicity [7,8]. Autophagy has been shown to have dual functions. Autophagy can enhance cellular function and survival by degrading damaged or unwanted organelles and by inhibiting apoptosis. Alternatively, autophagy can also induce cell death [9]. Several studies have shown that DOX treatment affects autophagy in vitro and in vivo. Some have shown that DOX treatment increases autophagy and some have shown that DOX decreases autophagy [10]. Since autophagy has dual functions in the life and death of cardiomyocytes, several investigators have employed chemical means of manipulating autophagy to elucidate its role in DOX-induced cardiotoxicity. However, it is important to note that many of the most commonly used chemical modulators of autophagy have off-target effects to be considered when interpreting results [11].

With all of the above molecular mechanisms leading to DOX-induced cardiotoxicity, its clinical use is limited. In this present study, our attempt was to investigate the effect of BGP-15 on DOX-induced injury. BGP-15 (O-[3-piperidino-2-hydroxy-1-propyl]-nicotinic amidoxime) possesses a wide range of cytoprotective effects but lacks a clear intracellular molecular target [12]. BGP-15 protects the mitochondrial membrane system, decreases oxidative stress [13], inhibits the nuclear translocation of apoptosis-inducing factor (AIF) from mitochondria, and inhibits mitogen-activated protein kinase (MAPK) activation [12]. BGP-15 shows several beneficial cardiovascular effects and has increasingly raised scientific interest in a wide range of pathological conditions in several disease models [14,15]. Although several protective mechanisms of BGP-15 were identified, its effects on DOX-induced cardiotoxicity are not yet investigated. In the current study, we have tested the effect of BGP-15 treatment on DOX-induced injury in H9c2 cells.

## 2. Results

### 2.1. Effects of BGP-15 Pretreatment on Cell Viability and LDH Release of DOX-Induced Cardiotoxicity

In order to evaluate the potential cardioprotective effect of BGP-15 against DOX-induced toxicity, a cell viability assay was carried out. As shown in Figure 1, panels A and B, 12 or 24 h of DOX exposure at doses between 0.1 and 3 μM induced a significant dose-dependent decrease in cell viability in comparison with the control group (*p* < 0.0001). Of note, no cytotoxicity was observed in response to 50 μM of BGP-15 alone. Thus, we assessed the effect of DOX on cell viability in the presence of BGP-15. Our findings showed that BGP-15 pretreatment at 0.1, 1, and 3 μM DOX groups significantly ameliorated cell viability in comparison with only DOX-treated cardiomyocytes at the same concentration after 12 h and 24 h of DOX exposure, respectively. 

To further confirm the protective effect of BGP-15 treatment on DOX-induced toxicity, the LDH content of cell culture media was determined by a colorimetric assay. Our results showed (Figure 1C) that DOX increased the LDH release of the cells in a dose-dependent manner. In line with the MTT assay, BGP-15 pretreatment did not just improve the cell viability, but also significantly decreased the DOX-induced LDH release of the cardiomyocytes. We measured a significant decrement in LDH release in the presence of BGP-15 compared to DOX treatment alone (1 μM DOX: 22.88 ± 0.78% vs. 17.04 ± 0.59% and 3 μM DOX: 25.91 ± 1.14% vs. 20.33 ± 0.85%).

### 2.2. BGP-15 Attenuates the DOX-Induced Generation of Mitochondrial ROS and Slightly Diminishes the Activation of Caspase-3 Apoptosis Marker in H9c2 Cells

ROS are involved in DOX-induced cell death [3]. Several studies have suggested that cardiomyocyte mitochondria are important intracellular targets of excess ROS during DOX-induced cardiotoxicity. Superoxide is one of the major ROS generated after DOX treatment [16]. Thus, to study the role of ROS in the protection induced by BGP-15 treatment (Figure 1), cells were analyzed for mitochondrial superoxide anion generation by flow cytometry in the presence or absence of BGP-15 in cardiomyocytes challenged by DOX treatment (Figure 2A). Our results indicated that DOX increased the mitochondrial superoxide generation compared to the control cells in a dose-dependent manner. Quantitative measurements of the mean fluorescence intensities of the samples demonstrated that 1 and 3 μM DOX alone significantly increased the ROS level (696.58 ± 42.34 and 992.03 ± 143.17, respectively) in contrast to the control group (408.18 ± 9.75). Conversely, enhanced MitoSOX fluorescence intensity induced by the DOX treatment was lessened by pretreatment with BGP-15, which was significantly lower in the BGP-15 + DOX3 group in comparison with DOX3-treated cells, indicating that the level of mitochondrial superoxide generation decreased in H9c2 cells in the presence of BGP-15. Fluorescent microscopy was employed to visualize MitoSOX staining (Figure 2B). However, we observed a notable accumulation of DOX in the nucleus, which makes it difficult to quantify the fluorescence intensity of microscopic images. These results suggest that decreased ROS generation may play a role in the cytoprotective effect of BGP-15 in H9c2 cells against DOX-induced cell toxicity.

In order to investigate the activation of apoptosis, we analyzed the ratio of cleaved-caspase-3 (17 kDa) /pro-caspase-3 (35 kDa) after the cardiomyocyte cells were exposed to 1 μM DOX for 24 h in the absence or presence of 50 μM BGP-15 pretreatment (Figure 2C). Our results showed that 1 μM DOX for 24 h significantly enhanced the ratio of cleaved-caspase-3/pro-caspase-3 (0.57 ± 0.08) in comparison with the control group (0.05 ± 0.01), indicating the activation of apoptosis. BGP-15 alone did not alter the ratio of cleaved-caspase-3/pro-caspase-3. Although the pretreatment of BGP-15 could slightly withhold the activation of apoptosis, unfortunately, the ratio of these abovementioned proteins was not statistically significant (0.43 ± 0.07) (*p* value = 0.26).

### 2.3. Effects of BGP-15 Pretreatment on Mitochondrial Depolarization of DOX-Exposed H9c2 Cells

Mitochondria are the primary target organelles of DOX-induced cardiotoxicity [17]. Mitochondrial membrane potential (MPP) is necessary for the production of ATP, which is crucial in living cells. JC-1 was used to assess the Δψ_m_ in H9c2 cardiomyocytes. This dye can selectively enter the mitochondria where it reversibly changes color as membrane potentials increase (over values of about 80–100 mV). The monomeric form of JC-1 in the cytosol emits a green fluorescence, and aggregates of the dye in the mitochondria of normal cells emit a red fluorescence. To confirm our fluorescent intensity (ratio red/green) results, JC-1 staining was carried out. Samples were visualized by fluorescent microscopy, with healthy mitochondria in red and unhealthy mitochondria in green. As shown in Figure 3A, B, the Dox-induced depolarization was mitigated by BGP-15 pretreatment. The ratio of fluorescent intensity was 120 ± 5.58 in the BGP-15 alone group. Our results revealed that DOX induced significant MMP loss in the 1 and 3 μM DOX groups (62.76 ± 4.36 and 50.43 ± 5.01, respectively) versus the control group (100%); however, MMP was recovered by the BGP-15 pretreatment (79.6 ± 5.75 and 63.94 ± 7.37, respectively), which was a significant improvement on BGP-15 + DOX 1 vs. DOX 1.

### 2.4. Effects of BGP-15 on Autophagy Flux in DOX-Induced Cytotoxicity

To monitor autophagic flux, cells were treated with chloroquine, which is a known autophagic flux inhibitor. Protein expression levels of LC3B (Figure 4A) and p62 (Figure 4C) were measured with Western blot, and lysosome and LC3B or p62 colocalization (Figure 4B,D) were determined by fluorescent microscopy. Our results showed that chloroquine significantly increased the LC3B relative protein expression in control vs. control+ chloroquine (1 ± 0 vs. 1.98 ± 0.23) and BGP-15 vs. BGP-15+ chloroquine (1.22 ± 0.08 vs. 2.37 ± 0.22) groups. However, chloroquine enhanced only moderately the LC3B expression in the case of DOX 1 (1.05 ± 0.11 vs. 1.59 ± 0.23) and BGP-15 + DOX 1 (1.02 ± 0.15 vs. 1.40 ± 0.18) groups. In contrast, expression of p62 was significantly reduced in the DOX 1 (0.16 ± 0.02) and BGP-15 + DOX 1 (0.11 ± 0.03) groups compared to the control (1 ± 0) and BGP-15 (1.12 ± 0.09) groups. Western-blot results were supported by microscopic images. However, it appears that modulation of autophagic flux is not likely to play a direct role in the cytoprotective effects of BGP-15 in DOX-induced toxicity.

## 3. Discussion

Pharmacological interventions that are able to enhance the resistance of myocardium against DOX-induced cardiac complications may offer a new perspective on the application of DOX in different tumors. In the current study, we found that BGP-15 mitigates DOX-induced cell death in H9c2 cells, evidenced by enhanced cell survival and decreased LDH release upon DOX treatment. Furthermore, BGP-15 decreased mitochondrial ROS production and mitochondrial depolarization in DOX-challenged cells. Earlier, BGP-15, a nicotinic acid derivative, has been shown to protect the myocardium against different injuries, including ischemia/reperfusion and heart failure with different triggers [14,15,18,19].

Mitochondrial dysfunction plays an important role in different cardiovascular diseases including DOX-induced cardiotoxicity. However, the mechanisms contributing to DOX-induced cardiotoxicity are not fully understood; the role of increased ROS production and enhanced oxidative stress appears to be one of the major factors. An enhanced amount of ROS impairs redox balance causing DNA damage, lipid peroxidation, mitochondrial dysfunction, and dysregulation of autophagy and apoptosis [20,21,22]. Ultimately, these alterations led to contractile dysfunctions, cardiomyopathy, and heart failure. DOX redox cycles on mitochondrial complex I, leading to ROS generation [23]. Moreover, an enhanced mitochondrial iron level upon DOX treatment also contributes to ROS generation [24]. Increased mitochondrial ROS leads to compromised mitochondrial integrity opening of the mitochondrial permeability transition pores, which causes modulation of Keap1/Nrf2 and alters regulation of the mitochondrial biogenesis [25,26]. BGP-15 has been shown to protect against oxidative stress and LPS-induced mitochondrial depolarization [27]. The authors suggested that BGP-15 inhibits mitochondrial Complex I and III, thereby suppressing ROS production and ultimately preventing the activation of ROS-dependent signaling pathways including MAPK and PARP and influencing cell death [27,28]. In line with the literature, we found decreased ROS production of cells treated with DOX in the presence of BGP-15, evidenced by the results of flow cytometry.

Moreover, BGP-15 prevented DOX-induced mitochondrial depolarization. Our results show a slight decrement in the activation of caspase-3 in the presence of BGP-15 in DOX-treated H9c2 cells. However, it was not statistically significant.

Impaired autophagy also plays a role in DOX-induced cell toxicity. Earlier, we have shown that DOX treatment impairs autophagic flux, which can be restored by metformin treatment. Metformin also targets mitochondrial complex I leading to a decreased ATP/AMP ratio, which activates AMPK and suppresses mTOR signaling leading to the activation of autophagy [29]. In the current study, we also found a weakened autophagy flux in the presence of DOX. Suppression of autophagic flux is mostly reported dose-dependently by DOX. It has been suggested that 1 μM DOX concentration does reflect the clinically relevant context [30], so we employed that concentration. However, BGP-15 did not restore it. Similar results were seen by Li et al. [31]. They observed that the treatment of NRVM with DOX (1 μM) resulted in a decrease in the autophagic flux within 6 h based on the measured LC3B-II and p62 levels. Moreover, by tracking lysosomes with Lysotracker Red, a fluorescence dye that labels acidic organelles, we also found that DOX decreased Lysotracker Red puncta. Although we did not quantify Lysotracker Red staining puncta, based on the microscopic results depicted in Figure 4, panels C and D, it is visible that upon DOX treatment the number of lysosomes decreased. It has been reported in some cell types that an increase in lysosome pH can impair the fusion of lysosome with autophagosomes [31,32]. Of note, based on our microscopic pictures, the fluorescent signals were slightly increased in BGP-15 + DOX1 + Q treated cells. Interestingly, the extent of autophagic flux perturbation correlated with the level of DOX-induced ROS production, leading further support to the notion that restoration of autophagic flux protects against DOX-induced cardiotoxicity. Taken together, based on our data, we cannot completely rule out that BGP-15 may influence autophagic flux; however, further studies need to be carried out to clarify the question.

In conclusion, our results indicated that BGP-15 could prevent DOX-induced cell toxicity by decreasing mitochondrial ROS production and attenuating mitochondrial depolarization.

## 4. Materials and Methods

### 4.1. Materials

Medium, serum, MTT, chloroquine, and LDH were purchased from Sigma (St. Louis, MO, USA). JC-1, MitoSOX and Lysotracker were bought from Life Technologies (Paisley, Scotland). Stain-Free gels and PVDF membrane were purchased from Bio-Rad Laboratories (Hercules, CA, USA). LC3B, p62, and Caspase-3 antibodies were obtained from Cell Signaling Technology (Boston, MA, USA). For fluorescent microscopy, p62 was purchased from Abcam (Cambridge, UK).

### 4.2. Cell Culture

The H9c2 cells were obtained from ATCC, CRL-1446, LGC Standards GmbH Wesel, Germany. Cells were maintained in Dulbecco’s modified Eagle’s medium (DMEM) supplemented with 10% fetal bovine serum and 1% penicillin-streptomycin at 37 °C in a humidified incubator consisting of 5% CO_2_ and 95% air. Cells were fed every 3 days, and cells were passaged by trypsinization when reaching 70–80% confluence. The passage number of the cells was between 8 and 26. The cells were pretreated with 50 μM BGP-15 for 24 h and then treated with DOX at indicated concentrations for 12 or 24 h. The stock solution of 2 mg/mL DOX was diluted weekly, and the BGP-15 solution was prepared before treatment in the medium with the composition mentioned above.

### 4.3. Cell Viability Assay by MTT

In this assay, cells were seeded into 96-well culture plates with 3000 cells/well, pretreated with 50 μM BGP-15 for 24 h, and treated with different doses (0.1; 1; 3 μM) of DOX for 12 or 24 h. After treatment, MTT solution (final concentration of 0.5 mg/mL) was added to each well and incubated for 3 h at 37 °C. After that, the medium was replaced by isopropyl alcohol to dissolve the formazan product. Absorbance was measured with a Multiskan GO Microplate Spectrophotometer (Thermo Fisher Scientific Oy, Ratastie, Finland) at 570 and 690 nm. The values were calculated as follows: the resulting colored solution is quantified by measuring absorbance at 570 nm and subtracting background absorbance at 690 nm. These values were expressed relative to the control, which was represented as 100% of viability. One percent H_2_O_2_ was used as the positive control. Absorbance values were averaged across 6 replicate wells and repeated 6–9 times.

### 4.4. Determination of Intracellular Reactive Oxygen Species Generation and Mitochondrial Function

In this experiment, 2000 cells/well were seeded on round glass coverslips placed into 24-well plates. Cells were treated with different doses (0.1; 1; 3 μM) of DOX for 24 h in the presence or absence of BGP-15 pretreatment (50 μM). At the end of the treatment, the medium was removed and cells were washed 3 times with Hank’s Balanced Salt Solution (HBSS). MitoSOX™ Red was added for 10 min at 37 °C in the dark. The nucleus was stained by DAPI. Finally, cells were fixed with 4% methanol-free formaldehyde and washed with HBSS, and coverslips were placed on a slide. Specimens were visualized using a fluorescence microscope. Images were captured by a Zeiss Axio Scope.A1 fluorescent microscope and analyzed with ZEN 2011 v.1.0.1.0. Software (Carl Zeiss Microscopy GmbH, München, Germany). The images were captured using the 63× oil immersion objective lens. For flow cytometry experiments, 20000 cells/well were seeded into 24-well plates, and the same protocol was carried out. Cells were trypsinized and fixed with 4% methanol-free formaldehyde. Cellular fluorescence was analyzed by a Guava Easy Cyte 6HT-2L flow cytometer (Merck Ltd., Darmstadt, Germany). MitoSOX Red was analyzed by using 510 nm excitation and 580 nm emission wavelengths. Using flow cytometry of H9c2 stained cells with and without MitoSOX Red, we were able to separate the red fluorescence signal elicited by DOX.

### 4.5. Assessment of Mitochondrial Membrane Potential

Mitochondrial membrane potential (MMP) was assessed using the fluorescent indicator 5,5’,6,6’-tetrachloro-1,1’3,3’-tetraethylbenzimidazolocarbo-cyanine iodide (JC-1; from Life Technologies (Paisley, Scotland)). Cells were seeded into black 96-well culture plates with 3000 cells/well and 24-well culture plates with coveslips (2000 cells/well), then non-pretreated/pretreated with 50 μM BGP-15, and treated with different doses (0.1; 1; 3 μM) of DOX for 24 h. After treatment, cells were incubated with 1 mg/mL JC-1 in Krebs–Henseleit buffer for 30 min at 37 °C. After incubation time, cells were washed once with Krebs–Henseleit buffer. Red and green fluorescence intensities of the samples were measured with a Multiskan GO Microplate Spectrophotometer (Thermo Fisher Scientific Oy, Ratastie, Finland) at 492 nm excitation and 520 and 590 nm emission wavelengths. DAPI as a nuclear stain was measured at 365 nm excitation and 445 emission wavelengths. The ratio of red and green fluorescence values was normalized to the blue fluorescence values. Using a spectrophotometer on H9c2-stained cells with and without JC-1 we were able to separate the red fluorescence signal elicited by DOX. Absorbance values were averaged across 4 replicate wells and repeated 5 times.

The coverslips were placed on a slide and visualized using a fluorescence microscope. Images were captured by Zeiss Axio Scope. A1 fluorescent microscope using the 63× oil immersion objective lens and analyzed with ZEN 2011 v.1.0.1.0. Software (Carl Zeiss Microscopy GmbH, München, Germany). A shift from red to green fluorescence indicates a loss of MMP, which was assessed by obtaining multiple merged images.

### 4.6. LDH (Lactate Dehydrogenase) Release Assay

LDH release was measured by the LDH-cytotoxicity assay kit (Sigma, St. Louis, MO, USA) according to the manufacturer’s instructions. Cells were seeded into 96-well culture plates with 5000 cells/well, pretreated with 50 μM BGP-15, and treated with different doses (0.1; 1; 3 μM) of DOX for 24 h. Absorbance was measured with a Multiskan GO Microplate Spectrophotometer (Thermo Fisher Scientific Oy, Ratastie, Finland) at 492 and 620 nm. The values were expressed relative to the positive control (2% TritonX-100 in assay medium), which was represented as maximal LDH release. Absorbance values were averaged across 8 replicate wells and repeated 5 times.

### 4.7. Autophagy Flux Determined by Fluorescent Microscopy

For the analyses of autophagy flux, we used Lysotracker Red, LC3B, and p62 antibodies. For these experiments, 2000 cells/well were seeded on round glass coverslips placed into 24-well culture plates. The treatment protocol was the following: with or without pretreatment with 50 μM BGP-15 and treated with different doses (1 μM) of DOX for 24 h, Rapamycin (5 mM) was used as the positive control, and the autophagic process was inhibited by chloroquine (10 mM, for 18 h). After treatments, the medium was removed, and cells were washed 3 times with HBSS. Lysotracker Red was added for 30 min at 37 °C in the dark. Cells were fixed with 4% methanol-free formaldehyde. Cells on coverlips were permeabilized and blocked with HBSS containing 5% normal goat serum and 0.3% TritonX-100 for 30 min. Thereafter, the cells were incubated with primary antibodies (LC3B or p62: 1:1000 with 1% BSA and 0.3% TritonX-100 in HBSS) for 2 h at 37 °C and incubated with a secondary antibody (Alexa Flour 488 goat anti-rabbit IgG (H + L) 1:500 with 0.2% BSA in HBSS) for 1 h at 37 °C in dark. The nucleus was stained by DAPI. The cells were washed with HBSS after each step. The coverslips were placed on a slide and visualized using a fluorescence microscope. Images were captured by a Zeiss Axio Scope. A1 fluorescent microscope and analyzed with ZEN 2011 v.1.0.1.0. Software (Carl Zeiss Microscopy GmbH, München, Germany). The images were captured using the 63× oil immersion objective lens.

### 4.8. Protein Isolation

After treatment, total protein fractions were extracted from the cultured H9C2 cells based on the previously described [33]. Afterward, the isolation protein concentration was determined using a BCA kit (Thermo Scientific, Rockford, IL, USA).

### 4.9. Western Blot Analysis

A 25 μg protein sample was loaded and separated in 4–20% Mini-PROTEAN^®^ TGX Stain-Free™ Protein gel. Then, gels were exposed to UV light, and thereby trihalo compounds contained in stain-free gels covalently bind to tryptophan residues in proteins, allowing total protein quantification, and were transferred onto PVDF membranes for 1 h at 100 V. Membranes were exposed by another brief irradiation, the resulting fluorescence signals were recorded, and the signal intensity was considered proportional to the total protein volume. After blocking with 5% of non-fat dry milk in Tris Buffered Saline with Tween 20 (TBST), membranes were incubated with primary antibody solution (LC3B, p62 and Caspase-3: 1:1000 in TBST) at 4 °C overnight. The membranes were washed with TBST and incubated with HRP-conjugated secondary antibody solution (1:3000 in TBST). After washing, the membranes were incubated with Clarity Western ECL substrate (Bio-Rad Laboratories) for visualization by enhanced chemiluminescence bands according to the recommended procedure (ChemiDoc Touch, Bio-Rad Laboratories). The chemiluminescent bands and each total protein lane intensity were measured by Image Lab software (version 5.2.1) (Bio-Rad Laboratories). During quantification, protein density is measured directly on the membranes and reflected in total loaded proteins. Thus, this type of normalization eliminates the need to select housekeeping proteins. The software calculates the normalization factor, which is the total volume (intensity) of the stain-free reference lane/total lane stain-free (intensity) of each lane. The protein expression was quantified by normalized volume, which means normalization factor x volume (intensity) [34].

### 4.10. Statistical Analysis

The data were expressed as mean ± SEM. Statistical analyses were performed with GraphPad Prism version 5 (La Jolla, CA, USA). The one-way analysis of variance (ANOVA) test was followed by Tukey’s multiple comparison tests, which identified the significant difference between control and treated groups, and the Šidák method was used to compare the treated groups (MitoSOX assay). A probability value of *p* < 0.05 was used as the criterion for statistical significance. Significant (*p* < 0.05), *, **, ***, and **** represent *p* < 0.05, *p* < 0.01, *p* < 0.001, and *p* < 0.0001 in the Tukey’s post-test, respectively.

## Figures and Tables

**Figure 1 ijms-24-05269-f001:**
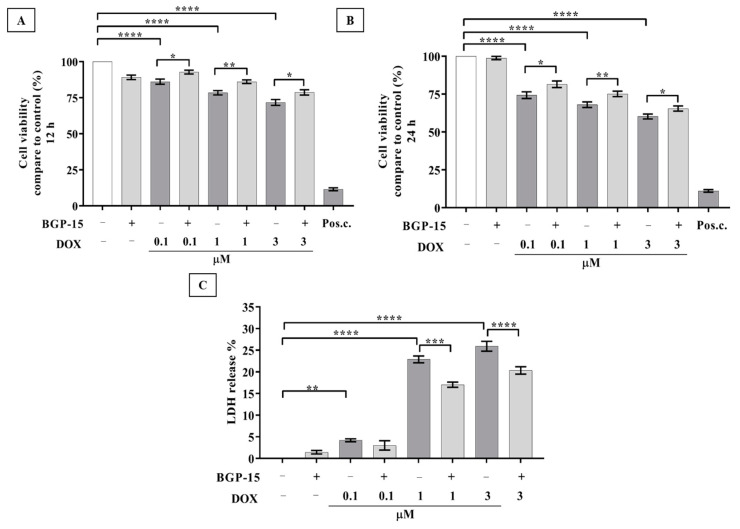
Effect of BGP-15 pretreatment on cell viability and LDH release in DOX-induced cardiotoxicity model. (**A**) Cell viability effects of BGP-15 (with or without 50 μM, 24 h pretreatment) on cardiomyocytes after 12 h (*n* = 6) and (**B**) 24 h (*n* = 8) incubation in the concentration range of 0.1–3 μM DOX. (**C**) LDH release % of BGP-15 (with or without 50 μM, 24 h pretreatment) on cardiomyocytes with 24 h incubation in the concentration range of 0.1–3 μM DOX, *n* = 5. Data are presented as mean ± SEM, *, **, ***, and **** represent *p* < 0.05, *p* < 0.01, *p* < 0.001 and *p* < 0.0001.

**Figure 2 ijms-24-05269-f002:**
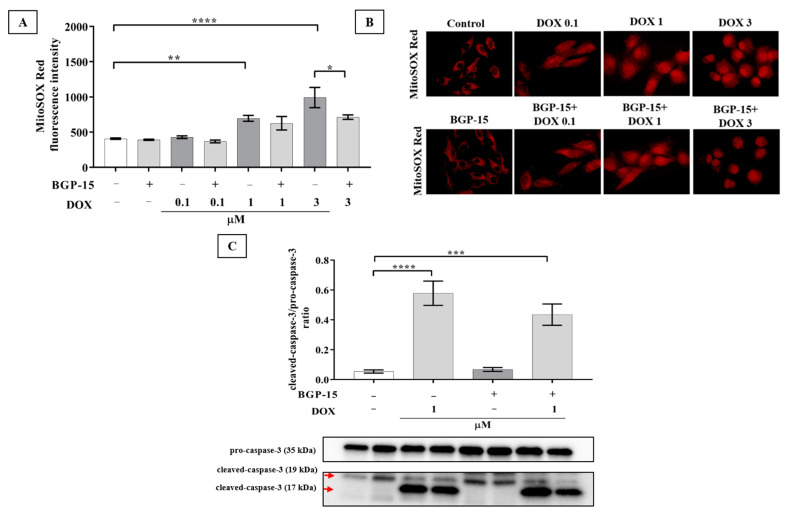
Effect of BGP-15 pretreatment on mitochondrial ROS generation and activation of caspase-3 apoptosis marker in DOX-induced cardiotoxicity. (**A**) ROS production was measured after MitoSOX Red staining by flow cytometry and expressed as mean ± SEM of MitoSOX Red fluorescence intensity (with or without BGP-15 (50 μM, 24 h pretreatment) on DOX-exposed H9c2 cells in the concentration range of 0.1–3 μM). *n* = 4. The images were captured using a 63× oil immersion objective lens. Data are presented as mean ± SEM, *, **, and **** represent *p* < 0.05, *p* < 0.01, and *p* < 0.0001, respectively. The significance of differences among groups was evaluated with a one-way analysis of variance (ANOVA) followed by Tukey’s posttest. (**B**) Representative images of MitoSOX Red staining; the nuclear fluorescence in DOX-treated H9c2 derives from DOX. (**C**) Analysis of the protein level of cleaved-caspase-3/pro-caspase-3 ratio after the cardiomyocyte cells were exposed to 1 μM DOX for 24 h in the absence or presence of the pretreatment of 50 μM BGP-15 for 24 h by Western blot. Red arrows indicate the bands for cleaced-caspase-3 (17; 19 kDa). Values were normalized to the total protein level and expressed as the mean ± SEM, *n* = 9. *** *p* < 0.001; **** *p* < 0.0001, respectively. The significance of differences among groups was evaluated with a one-way analysis of variance (ANOVA) followed by Tukey’s posttest.

**Figure 3 ijms-24-05269-f003:**
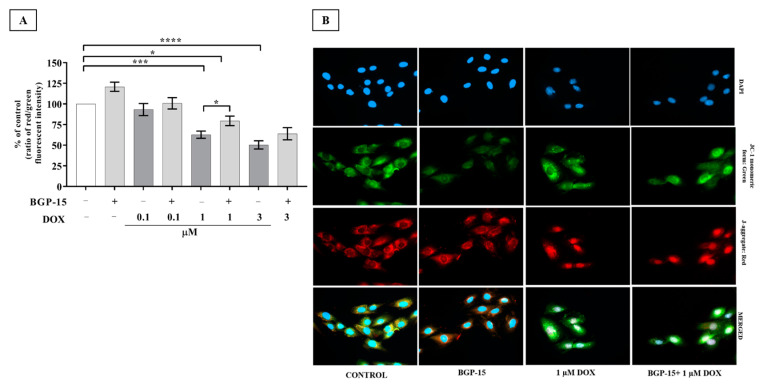
BGP-15 preserves the mitochondrial membrane potential in DOX-induced cardiotoxicity. (**A**) Effect of BGP-15 on DOX-induced mitochondrial membrane depolarization in H9c2 cells. Cells were exposed to 0.1, 1, and 3 μM DOX for 24 h in the absence or presence of 50 μM BGP-15 pretreatment for 24 h, then stained with JC-1, a membrane potential-sensitive fluorescent dye. Data are presented as mean ± SEM of % of control and ratio of red/green fluorescence intensity. *n* = 5. Data are presented as mean ± SEM, *, ***, and **** represent *p* < 0.05, *p* < 0.001, and *p* < 0.0001, respectively. (**B**) Representative images of JC-1 staining. Green channel: JC-1 monomeric form; Red channel: JC-1 aggregated form; Blue channel: DAPI as nucleus staining (the nuclear fluorescence in DOX-treated H9c2 derives from DOX); Merged images. The images were captured using a 63× oil immersion objective lens.

**Figure 4 ijms-24-05269-f004:**
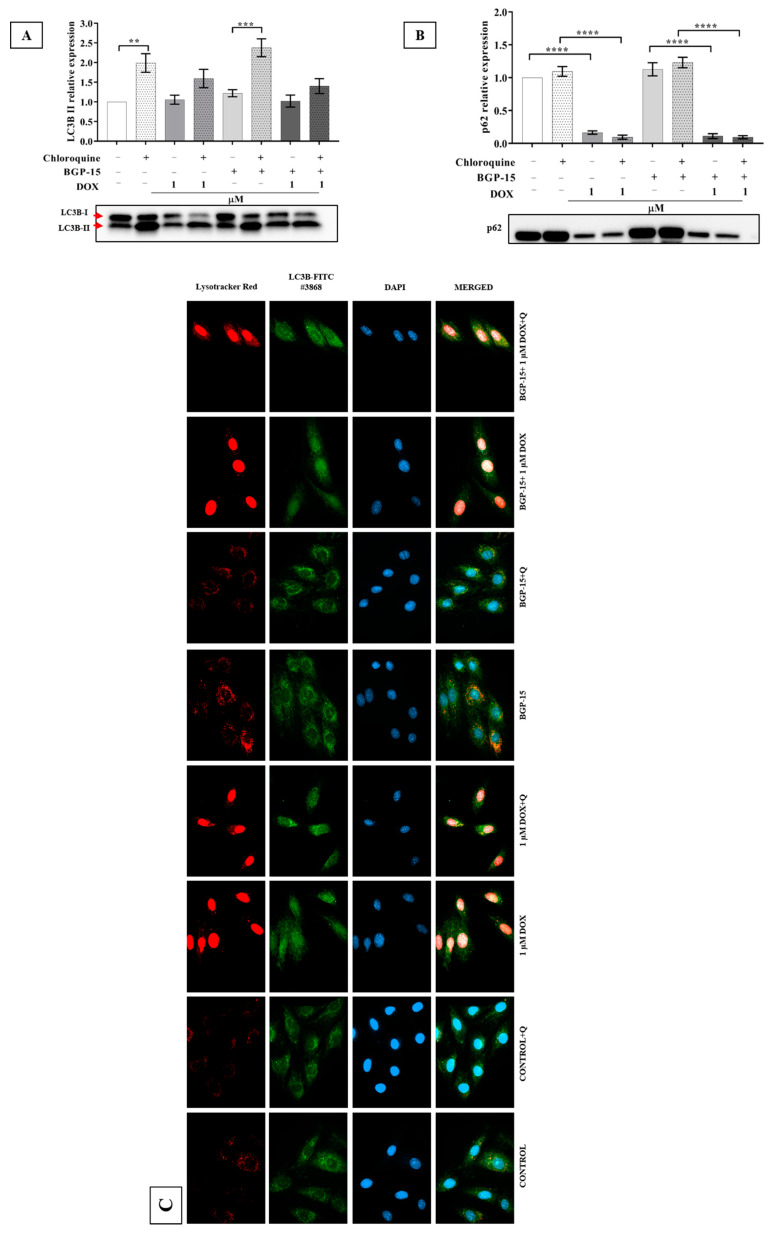
Effect of BGP-15 pretreatment and DOX-induced cardiotoxicity on autophagy flux. (**A**) Analysis of protein level of LC3B-II (**B**) and p62 after the cardiomyocyte cells were exposed to 1 μM DOX for 24 h in the absence or presence of the pretreatment of 50 μM BGP-15 for 24 h by Western blot. Values were normalized to the total protein level and expressed as the mean ± SEM, *n* = 14 and 14. The significance of differences among groups was evaluated with a one-way analysis of variance (ANOVA) followed by Tukey’s posttest. ** *p* < 0.01; *** *p* < 0.001; **** *p* < 0.0001, respectively. Red arrows indicate the bands for LC3B-I and LC3B-II. (**C**) Autophagy flux determined by fluorescent microscopy of LC3B or (**D**) p62 immunostaining. Cells were exposed to 1 μM DOX for 24 h in the absence or presence of 50 μM BGP-15 pretreatment for 24 h, and 10 μM chloroquine for 18 h, then stained. Representative images show the following: Blue channel: DAPI as nucleus staining (the nuclear fluorescence in DOX-treated H9c2 derives from DOX); Red channel: Lysotracker Red; Green channel: LC3B or p62 immunostaining; Merged images. The images were captured using a 63× oil immersion objective lens.

## Data Availability

All data used to support the findings of this study are available from the first author upon reasonable request.

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
