# Peer review of "BGP-15 Protects against Doxorubicin-Induced Cell Toxicity via Enhanced Mitochondrial Function"

_ijms, 2023, doi:10.3390/ijms24065269_

Round 1
Reviewer 1 Report
|
Date: |
February 12, 2023, |
|
|
|
|
Subject: |
Reviewer comments |
|
Manuscript ID.: ijms-2203941 Line 44-45; These sentence needs to reframe for better clarity. Line 80- 83; Interpretation of Cell viability not well described. Line 86-87; These sentence needs to reframe for better clarity. Line 91; Section: 2.2, Fig 1& 2 was not effectively discussed. The figure and bar graph needs to describe in more specific and required intercorrelated interpretation with the other obtained results. Line 105: Spelling error “indeicateing”. Line 109: Figure 1E, results were not discussed properly. And the H9c2 cell staining for the MitoSox red is varied with control, Dox 0.1 μM and Dox 0.1 +BGP-15. This variation in the MitoSox red staining in the nucleus needs to explain and also in discussion section this needs to address. Line 135; Spelling error “cytometria”. Line 146; “Figure 2; BGP-15 is ameliorated the mitochondrial membrane potential and reduced the level of apoptosis in DOX-induced cardiotoxicity” required modification as MMP was protected and the image 2B needs to replace with better quality. Image 2C, From the Western blot data from original image which image was selected is unclear and mention dose description for the image in the article. The bar diagram needs to mention for all groups and need result & discussion explainations. Line 161; Results were discussed very shallow without proper interpretation. Can describe about Figure 2C more from the point of apoptosis and with interrelation with obtained data in separate paragraph. Line 168-170; “/” symbol needs to delete for Figure 3/A… Also, the fluorescent microscopy results were not properly explained in this section. Need better correlation. Figure 3B image of lysotracker red for Dox treatment with/ without BGP15 has much differed from the control and Chloroquine required explanation for the image results or change the image with better quality. The result needs to explain well from the point of autophagy. Line 184-202; In Figure 3 Legend, the treatment regime can explain once in a common not required for each 3a, 3b, 3c & 3d. Also, author needs to discuss much about the TPN method in discussion part, as it is not commonly used in many article with proper reference. Line 276; Author mentioned only Fresh medium. The medium details of cell maintenance and during drug treatment/pretreatment requires to explain with specifications about the media composition for better clarification. Line 285; The MTT absorbance “570 and 690 nm” required author approach towards the cell viability data. For example. Mention 690nm usage for the subtraction of background absorbance value of 570nm to get the cell viability data………. etc. Line 373; Spelling mistake “salin”. Line 374; Dilution of primary and secondary antibodies required.
|
|

Author Response
Reviewer I.
First, we would like to thank for the suggestions and comments raised by this Reviewer, we tried to incorporate all of them in the revised version of our manuscript. We believe that the comments improve the quality of our manuscript. We responded point by point to all suggestions:
Reviewer: And in this article around 10 older references cited needs to add most recent references.
- Thank you for this valuable suggestion, we removed the following citations (number 3, 5, 11, 23) and inserted most recent references.
Reviewer: Authors needs to address the comment before acceptance are as follows;
Line 16-18; These sentence needs to reframe for better clarity.
Thank you for this valuable suggestion, we reframed it as the follow:
Several mechanisms have been proposed to play a role in DOX-induced cardiotoxicity, such as free radical generation, oxidative stress, mitochondrial dysfunction, altered apoptosis, and autophagy dysregulation.
Reviewer: Line 44-45; These sentence needs to reframe for better clarity.
Thank you for this valuable suggestion, we reframed it as the follow:
The mechanisms of DOX induced toxicity have not been clearly elucidated, but are known to involve, at least in part, mitochondrial dysfunction, leading to an increased generation of intracellular ROS, oxidative stress and apoptosis [1, 2]. Thus, DOX cardiotoxicity is closely associated with mitochondrial injury, which is characterized by iron overload, an early loss of mitochondrial membrane potential (MMP) followed by dysregulation of mitochondrial quality control mechanism.
Reviewer: Line 80- 83; Interpretation of Cell viability not well described.
We have modified the section as follows:
Our findings showed that BGP-15 pretreatment at 0.1, 1, 3 µM DOX groups was significantly ameliorate cell viability in comparison with only DOX treated cardiomyocytes at the same concentration after 12h and 24h of DOX exposure respectively.
Reviewer: Line 86-87; These sentence needs to reframe for better clarity.
Thank you for this valuable suggestion, we reframed it as the follow:
In line with MTT assay, BGP-15 pretreatment is not just improved the cell viability, but also significantly decreased the DOX induced LDH release of the cardiomyocytes.
Reviewer: Line 91; Section: 2.2, Fig 1& 2 was not effectively discussed. The figure and bar graph needs to describe in more specific and required intercorrelated interpretation with the other obtained results.
Thank you for this valuable suggestion, we have reorganized the Figures.
We add the following descriptions to the 2.1, and 2.2 sections:
2.1:
Thus, we assessed the effect of DOX on cell viability in the presence of BGP-15. Our findings showed that BGP-15 pretreatment at 0.1, 1, 3 µM DOX groups was significantly ameliorate cell viability in comparison with only DOX treated cardiomyocytes at the same concentration after 12h and 24h of DOX exposure respectively.
To further confirm the protective effect of BGP-15 treatment on DOX induced toxicity, LDH content of cell culture media were determined by a colorimetric assay. Our results showed (Figure 1C) that DOX increased LDH release of the cells in a dose-dependent manner. In line with MTT assay, BGP-15 pretreatment is not just improved the cell viability, but also significantly decreased the DOX induced LDH release of the cardiomyocytes.
2.2:
Thus, to study the role of ROS in the protection induced by BGP-15 treatment (Figure 1), cells were analyzed for mitochondrial superoxide anion generation by flow cytometry in the presence or absence of BGP-15 in cardiomyocytes challenged by DOX treatment (Figure 2A).
Conversely, enhanced MitoSOX fluorescence intensity induced by the DOX-treatment was lessened by pretreatment with BGP-15, which were significantly lower in BGP-15+DOX3 group in comparison with DOX3 treated cells, indicating that the level of mitochondrial superoxide generation decreased in H9c2 cells in the presence of BGP-15. Fluorescent microscopy was employed to visualize MitoSOX staining (Figure 2B). However, we observed notable accumulation of DOX in the nucleus, which makes difficult to quantify the fluorescence intensity of microscopic images. These, results suggest that decreased ROS generation may play a role in cytoprotective effect of BGP-15 in H9c2 cells against DOX-induced cell toxicity.
Reviewer: Line 105: Spelling error “indeicateing”. –
It is corrected.
Reviewer: Line 109: Figure 1E, results were not discussed properly. And the H9c2 cell staining for the MitoSox red is varied with control, Dox 0.1 μM and Dox 0.1 +BGP-15. This variation in the MitoSox red staining in the nucleus needs to explain and also in discussion section this needs to address.
Doxorubicin tends to accumulate in the nucleus of different cells [3]. Moreover, the emission maximum of doxorubicin is close the emission wavelength of MitoSOX red. These make difficult to quantify the intensity of red emission with fluorescence microscope. Instead of that we have used flow cytometry to quantify the ROS generation. We have performed the same protocol in the presence or absence of MitoSOX staining, which allow us to obtained the signal originated from MitoSOX staining.
We add the following to the results section:
However, we observed notable accumulation of DOX in the nucleus, which makes difficult to quantify the fluorescence intensity of microscopic images.
Reviewer: Line 135; Spelling error “cytometria”.
It is corrected.
Reviewer: Line 146; “Figure 2; BGP-15 is ameliorated the mitochondrial membrane potential and reduced the level of apoptosis in DOX-induced cardiotoxicity” required modification as MMP was protected and the image 2B needs to replace with better quality.
We have reorganized the Figures. Fig.2B was transferred to Fig.3B. We improve the quality and modified the title as follows:
BGP-15 preserves the mitochondrial membrane potential in DOX-induced cardiotoxicity.
Reviewer: Image 2C, From the Western blot data from original image which image was selected is unclear and mention dose description for the image in the article. The bar diagram needs to mention for all groups and need result & discussion explainations.
Thank you for the suggestion. Figure and results of apoptosis was reframed and transferred to the following section:
2.2. BGP-15 attenuates the DOX-induced generation of mitochondrial ROS and slightly diminish the activation of caspase-3 apoptosis marker in H9c2 cells.
In order to investigate the activation of apoptosis we analyzed the ratio of cleaved-caspase-3 (17 kDa) /pro-caspase-3 (35 kDa) after the cardiomyocyte cells were exposed to 1 µM DOX for 24 hours in the absence or presence of 50 μM BGP-15 pretreatment (Figure 2C). Our results showed that 1 µM DOX for 24 hours significantly enhanced the ratio of cleaved-caspase-3/pro-caspase-3 (0.57±0.08) in comparison with control group (0.05±0.01), indicating the activation of apoptosis. BGP-15 alone did not alter the ratio of cleaved-caspase-3/pro-caspase-3. Although, the pretreatment of BGP-15 could slightly withhold the activation of apoptosis, unfortunately the ratio of these abovementioned proteins was not statistically significant (0.43±0.07) (p value = 0.26).
Reviewer: Line 161; Results were discussed very shallow without proper interpretation. Can describe about Figure 2C more from the point of apoptosis and with interrelation with obtained data in separate paragraph.
Thank you for your suggestion we modified the section please see:
In order to investigate the activation of apoptosis we analyzed the ratio of cleaved-caspase-3 (17 kDa) /pro-caspase-3 (35 kDa) after the cardiomyocyte cells were exposed to 1 µM DOX for 24 hours in the absence or presence of 50 μM BGP-15 pretreatment (Figure 2C). Our results showed that 1 µM DOX for 24 hours significantly enhanced the ratio of cleaved-caspase-3/pro-caspase-3 (0.57±0.08) in comparison with control group (0.05±0.01), indicating the activation of apoptosis. BGP-15 alone did not alter the ratio of cleaved-caspase-3/pro-caspase-3. Although, the pretreatment of BGP-15 could slightly withhold the activation of apoptosis, unfortunately the ratio of these abovementioned proteins was not statistically significant (0.43±0.07) (p value = 0.26).
Reviewer: Line 168-170; “/” symbol needs to delete for Figure 3/A… Also, the fluorescent microscopy results were not properly explained in this section. Need better correlation. Figure 3B image of lysotracker red for Dox treatment with/ without BGP15 has much differed from the control and Chloroquine required explanation for the image results or change the image with better quality. The result needs to explain well from the point of autophagy.
Thank you for the suggestion.
We have improved the Figure, and add the following:
Although, we did not quantify Lysotracker red staining puncta, but based on the microscopic results, depicted in Figure 4 panel C and D, it is visible that upon DOX treatment the number of lysosomes are decreased.
“Taken together, based on our data we cannot completely rule out that BGP-15 may influence autophagic flux; however, further studies need to be carried out to clarify the question.
In conclusion, our results indicated that BGP-15 could prevent DOX-induced cell toxicity by decreasing mitochondrial ROS production, and attenuating mitochondrial de-polarization.”
Reviewer: Line 184-202; In Figure 3 Legend, the treatment regime can explain once in a common not required for each 3a, 3b, 3c & 3d. –
Thank you for this valuable suggestion, we corrected it.
Reviewer: Also, author needs to discuss much about the TPN method in discussion part, as it is not commonly used in many article with proper reference.
Since, increasing evidence suggests that housekeeping genes may change after drug treatment or under disease states, causing protein quantification errors in Western blot. Therefore, during Western blot analysis the chemiluminescent signals were not normalized through immunodetection of an internal standard housekeeping gene instead of, Stain-free gels were used. This technique allows us to detect the total loaded protein. Thus, the normalization is made against total loaded protein content.
The following sentence and reference is inserted to the methods section: The chemiluminescent bands and each total protein lane intensity were measured by Image Lab software (Bio-Rad Laboratories). During quantification, protein density is measured directly on the membranes and reflected to total loaded proteins. Thus, this type of normalization eliminates the need to select housekeeping protein. The software calculates normalization factor, which is total volume (intensity) of stain-free reference lane / total lane stain-free (intensity) of each lane. The protein expression was quantified by normalized volume, which means normalization factor x volume (intensity) [4].
Reviewer: Line 276; Author mentioned only Fresh medium. The medium details of cell maintenance and during drug treatment/pretreatment requires to explain with specifications about the media composition for better clarification. –
Thank you for this valuable suggestion, we clarified it.
Reviewer: Line 285; The MTT absorbance “570 and 690 nm” required author approach towards the cell viability data. For example. Mention 690nm usage for the subtraction of background absorbance value of 570nm to get the cell viability data………. etc. –
Thank you for this valuable suggestion, we inserted it.
Reviewer: Line 373; Spelling mistake “salin”.
It is corrected.
Reviewer: Line 374; Dilution of primary and secondary antibodies required.
Thank you for this valuable suggestion, we added the applied dilutions.
Again thank you for the suggestion by this reviewer, which substantially improved the quality of our manuscript.
Reviewer 2 Report
The authors Gyongyosi et al., have submitted an original manuscript entitled "BGP-15 protects against doxorubicin-induced cell toxicity via enhanced mitochondrial function" that demonstrates the effect of BGP-15 pretreatment to alleviate the harmful effects of doxorubicin treatment. It shows the mechanism of BGP-15 pretreatment by reducing the levels of reactive oxygen species and modulating the levels of autophagy. Following are some of the major comments that need to be addressed by the authors:
The abstract of the article could be improved by better introducing the reason for testing BGP-15 as a pretreatment. Moreover, the results need to be discussed for the overall result rather than mentioning the specifics like concentration.
The statements on lines 38-43 appear repetitive. The authors could revise this.
On line 48, the authors describe the involvement of DOX in apoptosis. However, since the study does not carry out any assays in this direction, this mention isn't justified.
On line 93, the authors need to provide a citation for the statement.
The results of Figure 1D are not significant from the data included. Therefore, the authors need to justify their findings in this context. Also in Figure 1E, the representative images do not provide sufficient evidence of the description mentioned. The authors need to revise the representative images and calculate the intensity, if possible.
Figures 1 and 2 need to be reorganized depending on the description in sections 2.1 and 2.2.
The results in Figure 3 are not described to support the claims made by the authors. Also, the figures do not have enough statistical differences to help their claims. The authors need to revise these results.
Author Response
First, we would like to thank for the suggestions and comments raised by this Reviewer, we tried to incorporate all of them in the revised version of our manuscript. We believe that the comments improve the quality of our manuscript. We responded point by point to all suggestions:
Reviewer: The abstract of the article could be improved by better introducing the reason for testing BGP-15 as a pretreatment.
We have modified the abstract as the follows:
Several mechanisms have been proposed to play a role in DOX-induced cardiotoxicity, such as free radical generation, oxidative stress, mitochondrial dysfunction, altered apoptosis, and autophagy dysregulation. BGP-15 has a wide range of cytoprotective effects including mitochondrial protection but up to now there is no information about its any beneficial effects on DOX-induced cardiotoxicity.
Reviewer: Moreover, the results need to be discussed for the overall result rather than mentioning the specifics like concentration.
We have modified the results section.
Reviewer: The statements on lines 38-43 appear repetitive. The authors could revise this. –
It is corrected
Reviewer: On line 48, the authors describe the involvement of DOX in apoptosis. However, since the study does not carry out any assays in this direction, this mention isn't justified.
We have omitted the following sentence:
„Additionally, DOX activated apoptosis by intrinsic and extrinsic pathways.”
Reviewer: On line 93, the authors need to provide a citation for the statement. –
Thank you, we inserted citation.
The results of Figure 1D are not significant from the data included. Therefore, the authors need to justify their findings in this context. Also in Figure 1E, the representative images do not provide sufficient evidence of the description mentioned. The authors need to revise the representative images and calculate the intensity, if possible.
Thank you for your suggestion. We have re-evaluated the results of flow cytometry with One way ANOVA followed by Sidak post hoc test, and we found significant differences between DOX3 and BGP-15+DOX3 groups.
Doxorubicin tends to accumulate in the nucleus of different cells [3]. Moreover, the emission maximum of doxorubicin is close the emission wavelength of MitoSOX red. These make difficult to quantify the intensity of red emission with fluorescence microscope. Instead of that we have used flow cytometry to quantify the ROS generation. We have performed the same protocol in the presence or absence of MitoSOX staining, which allow us to obtained the signal originated from MitoSOX staining.
Reviewer: Figures 1 and 2 need to be reorganized depending on the description in sections 2.1 and 2.2.
We have reorganized Fig. 1 and 2. The part of the original Fig.1 panel D and E is moved to Fig. 2 panel A and B, and Fig.2C remain in Fig2. The original Fig.2 panel A and B. are transferred a new Figure 3. Finally, the original Fig.3 is re-numbered as Fig 4.
2.1. Effects of BGP-15 pretreatment on cell viability and LDH release of DOX-induced cardiotoxicity
In order to evaluate the potential cardioprotective effect of BGP-15 against DOX-induced toxicity cell viability assay was carried out. As it is shown in Fig. 1 panel A and B, 12 or 24 h of DOX exposure, doses between 0.1 and 3 µM, induced a significant dose-dependent decrease in cell viability in comparison with control group (p<0.0001). To note, no cytotoxicity was observed in response to 50 µM of BGP-15 alone. Thus, we assessed the effect of DOX on cell viability in the presence of BGP-15. Our findings showed that BGP-15 pretreatment at 0.1, 1, 3 µM DOX groups was significantly ameliorate cell viability in comparison with only DOX treated cardiomyocytes at the same concentration after 12h and 24h of DOX exposure respectively.
To further confirm the protective effect of BGP-15 treatment on DOX induced toxicity, LDH content of cell culture media were determined by a colorimetric assay. Our results showed (Figure 1C) that DOX increased LDH release of the cells in a dose-dependent manner. In line with MTT assay, BGP-15 pretreatment is not just improved the cell viability, but also significantly decreased the DOX induced LDH release of the cardiomyocytes. We measured a significant decrement in LDH release in the presence of BGP-15 compared to DOX alone treatment (1 µM DOX: 22.88±0.78% vs. 17.04±0.59% and 3 µM DOX: 25.91±1.14% vs. 20.33±0.85%).
BGP-15 attenuates the DOX-induced generation of mitochondrial ROS and slightly diminish the activation of caspase-3 apoptosis marker in H9c2 cells
ROS are involved in DOX-induced cell death [1]. Several studies have suggested that cardiomyocyte mitochondria are important intracellular targets of excess ROS during DOX-induced cardiotoxicity. Superoxide is one of the major ROS generated after DOX treatment [5]. Thus, to study the role of ROS in the protection induced by BGP-15 treatment (Figure 1), cells were analyzed for mitochondrial superoxide anion generation by flow cytometry in the presence or absence of BGP-15 in cardiomyocytes challenged by DOX treatment (Figure 2A). Our results indicated that DOX increased the mitochondrial superoxide generation compared to the control cells in a dose-dependent manner. Quantitative measurements of the mean fluorescence intensities of the samples demonstrated that 1 and 3 µM DOX alone significantly increased the ROS level (696.58±42.34 and 992.03±143.17, respectively) in contrast to control group (408.18±9.75). Conversely, enhanced MitoSOX fluorescence intensity induced by the DOX-treatment was lessened by pretreatment with BGP-15, which were significantly lower in BGP-15+DOX3 group in comparison with DOX3 treated cells, indicating that the level of mitochondrial superoxide generation decreased in H9c2 cells in the presence of BGP-15. Fluorescent microscopy was employed to visualize MitoSOX staining (Figure 2B). However, we observed notable accumulation of DOX in the nucleus, which makes difficult to quantify the fluorescence intensity of microscopic images. These, results suggest that decreased ROS generation may play a role in cytoprotective effect of BGP-15 in H9c2 cells against DOX-induced cell toxicity.
In order to investigate the activation of apoptosis we analyzed the ratio of cleaved-caspase-3 (17 kDa) /pro-caspase-3 (35 kDa) after the cardiomyocyte cells were exposed to 1 µM DOX for 24 hours in the absence or presence of 50 μM BGP-15 pretreatment (Figure 2C). Our results showed that 1 µM DOX for 24 hours significantly enhanced the ratio of cleaved-caspase-3/pro-caspase-3 (0.57±0.08) in comparison with control group (0.05±0.01), indicating the activation of apoptosis. BGP-15 alone did not alter the ratio of cleaved-caspase-3/pro-caspase-3. Although, the pretreatment of BGP-15 could slightly withhold the activation of apoptosis, unfortunately the ratio of these abovementioned proteins was not statistically significant (0.43±0.07) (p value = 0.26).
Reviewer: The results in Figure 3 are not described to support the claims made by the authors. Also, the figures do not have enough statistical differences to help their claims. The authors need to revise these results.
We have changed the title of the section as the follows:
Effects of BGP-15 on autophagy flux in DOX-induced cytotoxicity.
And modified the discussion please see:
Although, we did not quantify lysotracker red staining puncta, but based on the microscopic results, depicted in Figure 4 panel C and D, it is visible that upon DOX treatment the number of lysosomes are decreased. It has been reported in some cell types that an in-crease in lysosome pH can impair the fusion of lysosome with autophagosomes [32]. Of note, based on our microscopic pictures, the fluorescent signals were slightly increased in BGP-15+DOX1+Q treated cells. Interestingly, the extent of autophagic flux perturbation correlated with the level of DOX-induced ROS production, leading further support to the notion that restoration of autophagic flux protects against DOX-induced cardiotoxicity. Taken together, based on our data we cannot completely rule out that BGP-15 may influ-ence autophagic flux; however, further studies need to be carried out to clarify the ques-tion.
In conclusion, our results indicated that BGP-15 could prevent DOX-induced cell toxicity by decreasing mitochondrial ROS production, and attenuating mitochondrial depolarization.
Again thank you for the suggestion by this reviewer, which substantially improved the quality of our manuscript.
Reviewer 3 Report
This study investigated whether BGP-15 can protect against doxorubicin-induced cardiotoxicity. The study identified that BGP-15 protected cardiac cells from doxorubicin-induced cardiotoxicity by reducing LDH release and apoptosis. It also reduced mitochondrial oxidative stress and prevented the loss of mitochondrial membrane potential, and modulated autophagic flux. These findings suggest that BGP-15 has the potential to alleviate DOX-induced cardiotoxicity, likely through its protective effects on mitochondria.
Author Response
We would like to thank the reviewer for his/her positive review.
Round 2
Reviewer 1 Report
No comments.